# The Costs of the COVID-19 on Subjective Well-Being: An Analysis of the Outbreak in Spain

**Ana Blasco-Belled \*, Claudia Tejada-Gallardo, Cristina Torrelles-Nadal and Carles Alsinet**

Department of Psychology, University of Lleida, 25001 Lleida, Spain; claudia.tejada@udl.cat (C.T.-G.); cristina.torrelles@udl.cat (C.T.-N.); carles.alsinet@udl.cat (C.A.)

**\*** Correspondence: ana.blasco@udl.cat

**Abstract:** The COVID-19 outbreak entailed radical shifts to individuals' daily habits that challenged their subjective well-being (SWB). Knowledge about the impact of COVID-19 on SWB is paramount for developing public policies to tackle mental health during health emergency periods. Decreases in life satisfaction are likely not only due to exposure to daily negative emotions but also due to hopelessness, fear, and avoidance of social interactions. We examined in a sample of 541 Spanish adults (1) reactions to the COVID-19 outbreak and (2) the mediating role of these reactions in the associations of hope and social phobia with life satisfaction through different levels of positive and negative affect. A moderated mediation analysis showed that the conditional indirect effect of hope and social anxiety on life satisfaction through information depended on the participants' having high positive affect and low negative affect. Affect seems to be a mechanism that modulates the influence of individuals' perception about COVID-19 on their life satisfaction. Those with high positive affect might see the "general picture" and search for adequate information as they avoid focusing on the problem and on specific information that precludes preventive behaviors. Having a positive affect might help individuals to adopt information-processing strategies during the COVID-19 outbreak that will improve their life satisfaction.

**Keywords:** coronavirus; COVID-19; mental health; subjective well-being; hope; social phobia

## 1. Introduction

On January 30, 2020, World Health Organization (WHO) declared an international public health emergency due to the novel COVID-19 outbreak [1]. The present pandemic originated in Wuhan, Hubei Province, China, in December 2019 when an initial cluster of pneumonia cases of unknown cause was associated with a common exposure: a seafood market in Wuhan [2]. Since the severe acute respiratory syndrome (SARS) outbreak in March 2003, no other global epidemic has appeared until COVID-19, which has turned out to be the largest outbreak of atypical pneumonia [3,4] and a threat to societal and individual sustainable development. The WHO confirmed 2,159,791 cases in the European region and, more specifically, 239,801 cases in Spain by the end of May [1]. This rapid escalation of cases led the Spanish government to consider restrictions in an effort to fight against the virus, but the official measures taken by governmental institutions to face the disease have been constantly changing. Because COVID-19 has rapidly spread worldwide through human-to-human transmission [5], situations involving social interactions can be seen as a threat, and individuals might try to avoid them. Additionally, the rapid increase of cases in a short-period time drove the Spanish government to take restrictive measures to prevent the transmission of the virus, including public measures to detect and control COVID-19 cases and to help citizens maintain physical distance from each other (e.g., closing of schools and universities, wearing masks, and keeping a minimum of one meter distance between individuals when outside the home) [6].

## 1.1. Mental Health During Pandemics

Pandemics and their subsequent containment measures (e.g., physical distancing) can have adverse effects on people's personal and social lives [4,7]. A positive mental health during a pandemic is a protective factor against future mental health diseases [3]. Subjective well-being (SWB) (i.e., high life satisfaction, high positive affect, and low negative affect) is considered an indicator of mental health [8], and its assessment is a key aspect in evaluating socio-psychosocial impacts within health emergency contexts. Lau et al. (2008) demonstrated that SWB remained fairly stable during the SARS pandemic within the Chinese population, reporting increases in community connectedness and future security. The experience of negative affect also appeared to be common. In fact, previous studies reported two specific aspects that contributed to negative mood and lower SWB in individuals facing the SARS pandemic [9]: the fear of being an agent of infection to others (especially family members) and fear of one's own mortality. Being preoccupied about one's own and one's peer's health and also about safety in the future is an important aspect that directly affects people's well-being [10] and can generate or exacerbate psychological distress during pandemics [11].

## 1.2. Can Hope and Social Phobia be Related to SWB During the COVID-19?

Life satisfaction captures individuals' evaluations of their life circumstances [12]; therefore, the context of a health crisis can yield to variations of these evaluations. It is important to understand the way individuals appraise these unprecedented circumstances and the potential effects on their SWB. A recent study showed that people's knowledge about COVID-19, their degree of confidence to overcome it, and their behavioral responses (such as avoiding crowded places) provided understanding about how people are dealing with COVID-19 [13]. Based on research, two psychological constructs can be relevant in studying the socio-psychological impact of COVID-19: hope and social phobia.

Hope is a state of mind that facilitates goal achievement and buffers against negative life events [14]. It generates means to overcome challenging circumstances [15], so it is considered a coping mechanism to increase life satisfaction [16]. On the other hand, social phobia is characterized by a persistent fear toward social scenarios, which implies the exposure to unknown individuals or situations [17,18] and can reduce quality of life [19]. People are more likely to suffer from social phobia when they experience negative feelings [7].

The results of the policy measures taken by the Spanish government at early stages of the outbreak (prior to the confinement) might have played an important role in their citizens' emotions and perceptions of the situation and the forthcoming restrictive measures. Considering that life satisfaction can be influenced by day-to-day emotions [20] and that positive affect (as opposed to negative affect) facilitates the development of personal and social coping resources [21], the relationships of hope and social phobia with life satisfaction could be conditioned to momentary states of positive or negative emotions during the outbreak. On this basis, we propose that individuals' appraisals about COVID-19 would mediate the association of hope and social phobia with their degree of life satisfaction (hope/social phobia → COVID-19 → life satisfaction), and this mediating effect would depend on the individuals' level of affect (hope/social phobia → COVID-19 → affect → life satisfaction).

## 1.3. Aims of the Study

Knowledge about the socio-psychological impact of the coronavirus is paramount for developing public policies aimed at improving a population's mental health during health emergency periods. Since the beginning of the COVID-19 emergency, an increasing number of studies are providing clues to understand and calculate the impact of this outbreak on mental health. However, no previous studies have accounted for the socio-psychological impact in Spain or accounted for beliefs of fear in the present (social phobia) and hope for the future. We attempt to investigate reactions to the COVID-19 outbreak, namely the degree of information that people have reported about the situation (i.e., information), the self-perceived degree of COVID-19 as a social

threat (i.e., threat), the fear of being infected (i.e., infection) and the agreement with the protective measures adopted by the Spanish government (i.e., measures). In efforts to examine COVID-19's impact on SWB, we analyzed the mediating role of the responses to these COVID-19 questions on the relationships of hope and social phobia with life satisfaction, and whether these relationships depended on specific levels of positive and negative affect—known as conditional indirect effects or moderated mediation effects.

We propose the following hypotheses, which are speculative in nature since no previous research has provided evidence with this respect:

**Hypothesis 1 (H1: simple mediation***): Responses to the COVID-19 questions (information, threat, infection, and measures) will mediate the effect of hope on life satisfaction.*

**Hypothesis 2 (H2: simple mediation):** *Responses to the COVID-19 questions (information, threat, infection, and measures) will mediate the effect of social phobia on life satisfaction.*

**Hypothesis 3 (H3: moderated mediation):** *The indirect effect of information, threat, infection, and measures on the hope–life satisfaction relationship will be moderated by different levels of (positive and negative) affect.*

**Hypothesis 4 (H4: moderated mediation):** *The indirect effect of information, threat, infection, and measures on the social phobia–life satisfaction relationship will be moderated by different levels of (positive and negative) affect.*

## 2. Materials and Methods

### 2.1. Participants and Procedure

A sample of 541 Spanish adults (*N* = 541) ranging from 18 to 74 years old (M = 38.82; SD = 15.97, 65.8% women) answered an anonymous 10-minute online survey, which they received via the snowball sampling method. Given the difficulty of collecting a community-based sampling, online data were obtained through our contact networks via WhatsApp. By this means, a participant could send the link of the survey to their acquaintances so that they could complete it and share it with their contact networks. We also posted an announcement about the study's aim, information about the study, and a link to the survey on the research group's Facebook page and Instagram profile. We developed a survey assessing demographics and the study questionnaires in a Google form so participants could easily access it through smartphones. Despite our initial focus of participants from Catalonia, respondents from other autonomous communities of Spain were also eligible. Participants had to provide informed consent, and they could withdraw from the study at any time. We collected data from March 12 to 15, 2020, a week before the Spanish government declared the state of alarm and implemented full restrictions. The study was approved by the data protection committee of the University.

### 2.2. Instruments

Sociodemographic data were collected on age, gender, marital status, employment situation, socio-economic status, and education level. To measure the socio-psychological impact of COVID-19, we used the following self-administered scales.

Life satisfaction was measured through the Personal Well-being Index (PWI) [22,23]. This scale includes seven items evaluating respondents' satisfaction with different domains: health, standard of living, personal safety, the community groups to which they belong, future security, and relationships with others. These were assessed on a 10-point Likert scale (0 = completely unsatisfactory, 10 = completely satisfactory). A sample item is "to what extent are you satisfied with your health?" The reliability estimate was good ($\alpha$ = 0.89).

Hope was measured by the Adult Dispositional Hope Scale (ADHS) [14,24], a 12-item questionnaire measuring the participants' level of hope. The responses were evaluated using an 8-

point Likert scale (1 = definitely false, 4 = definitely true). A sample item is "I energetically pursue my goals." The reliability of the scale was good (α = 0.87).

Social phobia was measured by the Liebowitz Social Anxiety Scale (LSAS-SR) [25,26], a scale that includes 24 items and evaluates anxiety (or fear) and avoidance of social interactions and performance situations. For each of the 24 items, respondents derived answers of anxiety and avoidance of given situations. Participants evaluated the subscales of anxiety and avoidance on a 3-point Likert scale (for anxiety: 0 = none, 3 = severe; for avoidance: 0 = never, 3 = usually). Sample items are "eating in public places" and "participating in small groups." The reliability estimates indicated excellent values for social anxiety (α = 0.94) and social avoidance (α = 0.92).

Positive and negative affect were measured by the Scale of Positive and Negative Experiences (SPANE) [27,28], a 12-item questionnaire that evaluates how much respondents have experienced certain feelings in the past week on a 5-point Likert scale (1 = very rarely or never, 5 = very often or always). A sample item is "pleasant feelings" for positive affect, and "unpleasant feelings" for negative affect. Reliability estimates were excellent for positive affect (α = 0.95) and good for negative affect (α = 0.88).

We also included the following four questions related to the COVID-19 outbreak, which respondents rated on a 10-point Likert scale: (1) What is your degree of knowledge about COVID-19? (1 = totally uninformed, 10 = totally informed), (2) To what extent do you believe that COVID-19 is a serious threat for public health? (1 = no threat, 10 = extremely serious threat) (3) To what extent do you think you can be infected by the COVID-19? (1 = no probability of infection, 10 = extremely high probability of infection), (4) Do you believe that the protective measures adopted by the Spanish government are adequate and proportionate? (1 = completely inadequate and disproportionate, 10 = completely adequate and proportionate).

## 2.3. Data Analysis

To test our hypothesis about the mediating role of the COVID-19 information, threat, infection, and measures, we constructed six conditional processes to assess moderated mediational analysis (i.e., how mediating variables are influenced by moderators). These analyses were carried out using the macro PROCESS for SPSS [29] and following the Preacher and Hayes (2008) procedure [30], which uses bias-corrected bootstrap estimates and 95% confidence intervals to infer specific and total indirect effects, assuming that normality distributions are rarely met. Bootstrapping is a non-parametrical procedure in which no assumptions about the shape of the sampling distribution of the statistic are necessary when conducting inferential tests. It provides standard errors that are robust to violations of normality [31], and it generates non-symmetric confidence intervals, which are particularly useful for parameter estimates based within non-normal sampling distributions (e.g., indirect effects) that make it suitable for the purpose of the present study [32]. Demographics were controlled for in all analyses. The data are available at a public OSF page: https://osf.io/ce2zf/?view_only=b1ca010d2bff42dcb6fae2049ccd08e0.

We first analyzed two simple mediation models to test whether the COVID-19 questions, and therefore self-reported judgments about how people understood and faced the outbreak, mediated the relationship of hope and social phobia (anxiety and avoidance) with life satisfaction, that is, whether the effect of stable judgments about life satisfaction on social phobia and hope will be influenced by people's evaluations of the COVID-19 outbreak. The two models were analyzed applying Model 4 in PROCESS, in which (1) hope and (2) social phobia were entered as predictors, the COVID-19 questions as mediators and life satisfaction as the criterion of the two different models, respectively. All variables were mean centered in the mediation analysis [33]. Figure 1 depicts the conceptual diagram of the simple mediation model, which was analyzed for the two predictors.

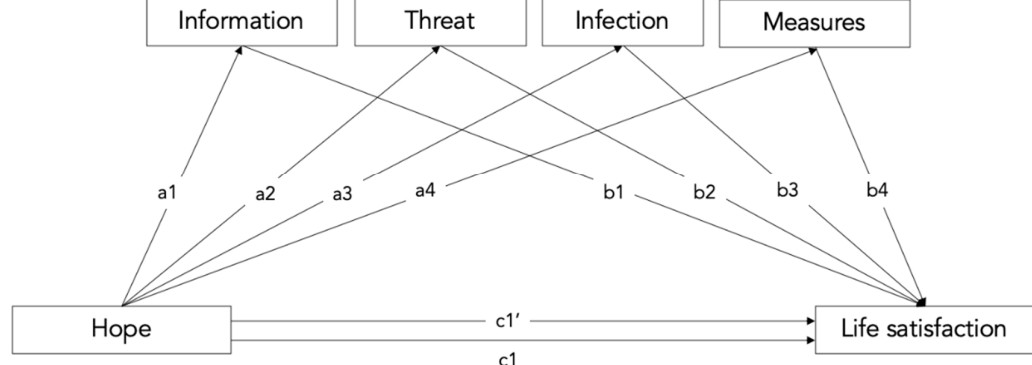

**Figure 1.** Conceptual diagram of a simple mediation model with the COVID-19 questions as mediators between hope and life satisfaction. The same model was analyzed with anxiety and avoidance (social phobia) as predictors.

As a second step, we tested whether the strength of the indirect effects of the COVID-19 questions (on life satisfaction) depended on the level of affect by means of moderated mediation models, which are recommended over simple mediated models [34,35]. This procedure allowed us to investigate the conditional direct and indirect effects of the mediation model (e.g., the conditions under which information, threat, infection, and measures mediate the relationships). We investigated the magnitude of the conditional indirect effects of hope and social phobia on life satisfaction via responses to the COVID-19 questions depending on high (+1SD), mean, and low (−1SD) levels of positive and negative affect. This informed us about "the how of the when" [36] (p.1)—that is, the mechanisms by which the predictors influence life satisfaction through information, threat, infection, and measures. Four different models were analyzed, in which the predictors (hope and social phobia), mediators (COVID-19 questions), and criterion (life satisfaction) remained the same, and we introduced two moderators (positive affect and negative affect) to see whether the immediate experiences of affect conditioned the meditations (Model 59 in PROCESS). To test the associations, an index of moderated mediation was obtained, which evaluates the link between the indirect effect (e.g., hope → information → life satisfaction) and the moderator (e.g., positive affect). If this index is different from zero, it can be taken as an instance of moderated mediation. Significant results are interpreted when the bootstrap 95% confidence interval is not included. Figure 2 shows a conceptual diagram of the moderated mediation model.

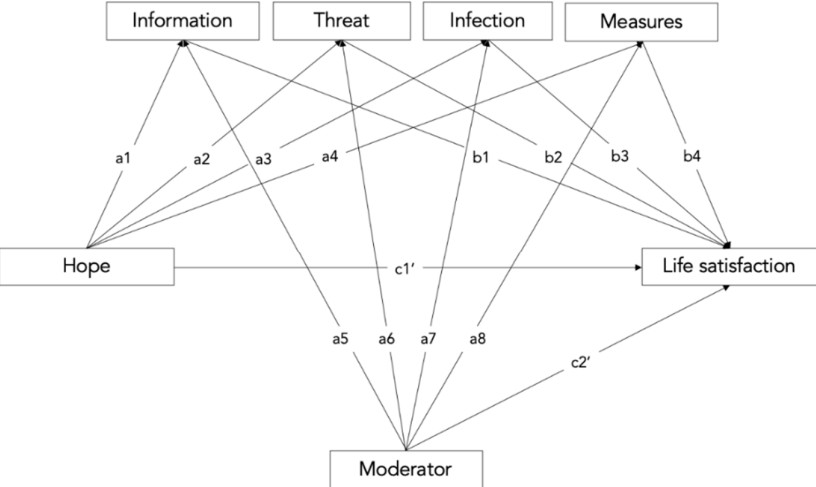

**Figure 2.** Conceptual diagram of a moderated mediation model for conditional direct and indirect effects. The same model was analyzed with anxiety and avoidance (social phobia) as predictors.

As an example, Figure 3 presents the statistical diagram with path coefficients of a simple mediation model testing H1. The estimated equation of the indirect effect of hope on life satisfaction through information was $\omega = a1b1$, while the direct effect of hope on life satisfaction was a function of c1'. Figure 4 presents a statistical diagram with path coefficients of moderated mediation testing H3. The estimated equation of the indirect effect of hope (X) on life satisfaction (Y) through information (M) depending on positive affect (W) was $\omega = (a1 + a9W)(b1 + b5)$, while the estimated equation of the conditional direct effect of hope (X) on life satisfaction (Y) was a function of c1' + c3'W.

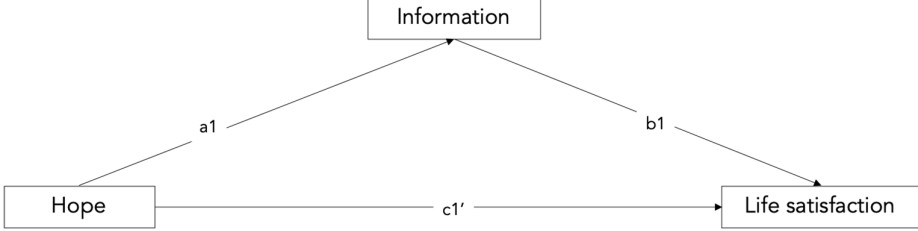

**Figure 3.** Statistical diagram of simple mediation model for direct and indirect effects for Hypothesis 1.

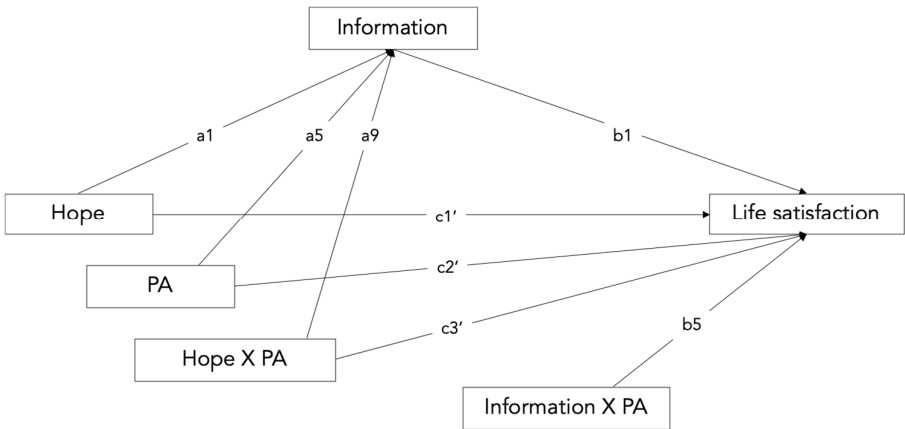

**Figure 4.** Statistical diagram of moderated mediation model for conditional direct and indirect effects for Hypothesis 3; PA = Positive affect.

## 3. Results

### 3.1. Demographics

Participants were from different regions of Spain, including Madrid, Basque Country, Aragon, Andalusia, Murcia, Valencia, Estremadura, Galicia, the Rioja region, and Castile-Leon, but mainly from Catalonia. They provided information about their marital status (49.4% single, 41.2% married, 7.6% separated or divorced, and 1.8% widowed), perceived socio-economic level (1.5% significantly higher than average, 6.3% higher than average, 20.5% slightly higher than average, 54.9% average, 10.4% slightly lower than average, 4.6% lower than average, and 1.8% significantly lower than average), working status (65.2% employed, 2.2% unemployed, 23.1% student, and 9.4% others), and level of education (1.3% primary education, 15.5% secondary education, 47.5% graduates (including university), 35.3% post-graduates (master and PhD), and 0.4% other).

### 3.2. Descriptive Statistics

Table 1 shows the descriptive statistics and correlations among the studied variables.

**Table 1.** Descriptive statistics, reliability estimates, and correlations of the analyzed variables.

| Scale | M(SD) | Min– Max | Correlations | | | | | | | | |
|---|---|---|---|---|---|---|---|---|---|---|---|
| | | | 1 | 2 | 3 | 4 | 5 | 6 | 7 | 8 | 9 |
| 1. Life satisfaction | 7.32(1.49) | 0–10 | | | | | | | | | |
| 2. Hope | 6.20(0.99) | 1.25–8 | 0.63** | | | | | | | | |
| 3. Social anxiety | 0.66(0.53) | 0–2.75 | −0.22** | −0.15** | | | | | | | |
| 4. Social avoid | 0.75(0.54) | 0–2.88 | −0.19** | −0.14** | 0.77** | | | | | | |
| 5. Positive affect | 3.33(1.05) | 1–5 | 0.21** | 0.17** | −0.19** | −0.15** | | | | | |
| 6. Negative affect | 2.70(0.97) | 1–5 | −0.25** | −0.13** | 0.27** | 0.22** | −0.67* | | | | |
| 7. Information | 7.06(1.93) | 1–10 | 0.24** | 0.19** | −0.10* | −0.93* | 0.01 | −0.06 | | | |
| 8. Threat | 6.80(2.11) | 1–10 | 0.02 | 0.05 | 0.10* | 0.12* | −0.10* | 0.21** | 0.21** | | |
| 9. Infection | 6.11(2.07) | 1–10 | 0.05 | 0.08 | 0.04 | 0.03 | −0.10* | 0.17** | 0.14** | 0.23** | |
| 10. Measures | 5.18(2.31) | 1–10 | 0.17** | 0.03 | −0.02 | 0.01 | 0.02 | −0.10* | 0.25** | 0.14** | −0.03 |

\* $p < 0.05$, \*\* $p < 0.005$; Note. Social avoid = social avoidance.

### 3.3. Simple Mediation Analysis

Responses to the COVID-19 questions were expected to mediate the relationship of (1) hope and (2) social phobia with life satisfaction (Model 4 in PROCESS). Path coefficients from the two models, based on Figure 1, are shown in Table 2.

**Table 2.** Path coefficients for the simple mediation models of hope and social phobia.

| Path | Model Hope | Model Social Phobia (Anxiety/Avoidance) |
|---|---|---|
| a1 | 0.36 ***(0.08) | −0.36 *(0.16)/−0.33 *(0.15) |
| a2 | 0.11(0.09) | 0.41 *(0.17)/0.46 **(0.17) |
| a3 | 0.17(0.09) | 0.15(0.17)/0.13(0.17) |
| a4 | 0.07(0.10) | −0.05(0.18)/0.05(0.18) |
| b1 | 0.08 **(0.03) | 0.15 ***(0.03)/0.15 ***(0.04) |
| b2 | −0.03(0.02) | −0.02(0.03)/−0.02(0.03) |
| b3 | −0.03(0.02) | 0.03(0.03)/0.02(0.03) |
| b4 | 0.08 ***(0.02) | 0.08 **(0.03)/0.08 **(0.03) |
| c1′ | 0.93 ***(0.05) | −0.56 ***(0.12)/−0.47 ***(0.12) |
| c1 | 0.95 ***(0.05) | −0.62 ***(0.12)/−0.51 ***(0.12) |

\* $p < 0.05$, \*\* $p < 0.01$, \*\*\*, $p < 0.001$.

Results from the hope model ($F_{(1-539)} = 19.25$, $R^2 = 0.04$, $p < 0.001$) demonstrated that the total effect (c1) on life satisfaction was significant (c1 = 0.954, SE (standard error) = 0.050, $t = 18.931$, $p < 0.001$), which was very similar to the direct effect (c1′) after controlling for the mediators' effect (c1′ = 0.923, SE = 0.050, $t = 18.407$, $p < 0.001$). The total indirect effect, accounted as the sum of specific indirect effects, was significant ($\beta = 0.031$, SE = 0.017, LL (lower limit) = 0.000, UL (upper limit) = 0.066), with only the specific indirect effects through information also being significant (a1b1 = 0.029, SE = 0.013, LL = 0.008, UL = 0.058). This suggests that the perceived degree of information about the COVID-19 situation influenced the effect of hope on life satisfaction. Regarding social phobia, in the anxiety model ($F_{(1-539)} = 5.48$, $R^2 = 0.01$, $p < 0.05$) the total effect (c1 = −0.617, SE = 0.118, $t = −5.241$, $p < 0.001$) and the direct effect were significant (c1′ = −0.558, SE = 0.116, $t = −4.812$, $p < 0.001$), whereas the total indirect effect was non-significant ($\beta = −0.060$, SE = 0.039, LL = −0.145, UL = 0.009). In the social avoidance model ($F_{(1-539)} = 4.66$, $R^2 = 0.01$, $p < 0.05$), the total effect (c1 = −0.515, SE = 0.117, $t = −4.409$, $p < 0.001$) and the direct effect (c1′ = −0.465, SE = 0.115, $t = −4.050$, $p < 0.001$) were also significant, and the total indirect effect was non-significant ($\beta = −0.049$, SE = 0.041, LL = −0.136, UL = 0.024). Specific indirect effects via information were significant in the social anxiety (a1b1 = −0.053, SE = 0.028, LL = −0.115, UL = −0.010) and the social avoidance models (a1b1 = −0.049, SE = 0.028, LL = −0.111, UL = −0.003). The path coefficient reported significant results for information, threat, and measures, suggesting that the relationship between social phobia and life satisfaction was also explained through these variables. Overall, these results partially supported Hypotheses 1 and 2.

### 3.4. Moderated Mediation Analysis

To examine whether different levels of transient affective states (positive and negative affect) can influence the conditional indirect effects of hope and social phobia on life satisfaction through information, threat, infection, and measures, we constructed six conditional processes (Model 58 of PROCESS) on 5000 bootstrapped samples. We analyzed a total of six moderated mediation models: (1) positive affect and (2) negative affect as moderator (hope model), (3) positive affect and (4) negative affect as moderators for social phobia–anxiety, and (5) positive affect and (6) negative affect as moderators for social phobia–avoidance. The path coefficients for hope and social phobia, based on Figure 2, are presented in Table 3. The indirect effects of simple mediation models for hope and social phobia are shown in Table 4.

**Table 3.** Path coefficients for the moderated mediation models of hope and social phobia.

| Path | Model Hope | Model Social Phobia (Anxiety/Avoidance) | Model Hope | Model Social Phobia (Anxiety/Avoidance) |
|---|---|---|---|---|
| | | **Moderator = Positive Affect** | **Moderator = Negative Affect** | |
| a1 | 0.37 **(0.08) | −0.41 *(0.16)/−0.34 *(0.16) | 0.37 ***(0.09) | −0.40 *(0.17)/−0.31(0.16) |
| a2 | 0.15(0.09) | 0.37 *(0.18)/0.43 *(0.17) | 0.14(0.09) | 0.29(0.18)/0.36 *(0.17) |
| a3 | 0.20 *(0.09) | 0.01(0.18)/0.04(0.17) | 0.20 *(0.09) | −0.09(0.18)/−0.06(0.17) |
| a4 | 0.06(0.10) | −0.14(0.20)/0.04(0.19) | 0.02(0.10) | 0.00(0.20)/0.31(0.19) |
| a5 | −0.03(0.08) | −0.03(0.08)/−0.01(0.08) | −0.07(0.09) | −0.06(0.09)/0.13(0.19) |
| a6 | −0.23 *(0.09) | −0.17(0.09)/−0.17 *(0.09) | 0.47 ***(0.09) | 0.40 ***(0.10)/0.41 ***(0.09) |
| a7 | −0.22 *(0.09) | −0.21 *(0.09)/−0.20*(0.09) | 0.38 ***(0.09) | 0.37 ***(0.09)/0.36 ***(0.09) |
| a8 | 0.04(0.10) | 0.01(0.10)/0.04(0.10) | −0.24 *(0.10) | −24 *(0.11)/−26 *(0.11) |
| a9 | −0.03(0.08) | −0.16(0.15)/−0.05(0.14) | −0.08(0.08) | 0.22(0.17)/0.03(0.16) |
| b1 | 0.07 *(0.03) | 0.14 ***(0.03)/0.15 *** (0.03) | 0.08 **(0.05) | 0.14 ***(0.03)/0.14 ***(0.03) |
| b2 | −0.05(0.02) | −0.01(0.03)/−0.01(0.03) | −0.02(0.03) | −01(0.03)/0.01(0.03) |
| b3 | 0.00(0.02) | 0.03(0.03)/0.04(0.03) | 0.01(0.02) | 0.04(0.03)/0.04(0.03) |
| b4 | 0.07 **(0.02) | 0.08 **(0.03)/0.08 **(0.03) | 0.07 **(0.02) | 0.06 *(0.03)/0.07 *(0.03) |
| b5 | −0.01(0.03) | −0.04(0.11)/0.11(0.11) | 0.14**(0.05) | 0.10(0.12)/0.00(0.12) |
| c1' | 0.89 ***(0.05) | −0.50 ***(0.12)/−0.38**(0.12) | 86 ***(0.05) | −0.46 ***(0.13)/−0.36 **(0.12) |
| c2' | 0.16 **(0.05) | 0.26 ***(0.06)/0.29 ***(0.06) | −0.24 ***(0.05) | −0.31 ***(0.07)/−0.33 ***(0.07) |
| c3' | −0.05(0.05) | −0.03(0.03)/−0.02(0.03) | −0.03(0.03) | −0.04(0.04)/−0.05(0.04) |

\* $p < 0.05$, ** $p < 0.01$, *** $p < 0.001$.

**Table 4.** Indirect effects of hope on life satisfaction via information, threat, infection and measures.

| | Outcome | | | | | | | | |
|---|---|---|---|---|---|---|---|---|---|
| | Hope | | | Social Anxiety | | | Social Avoidance | | |
| Mediator | Estimate (SE) | 95% CI | | Estimate (SE) | 95% CI | | Estimate (SE) | 95% CI | |
| | | LL | UL | | LL | UL | | LL | UL |
| Information | 0.03(0.01) | 0.007 | 0.057 | −0.05(0.03) | 0.114 | −0.008 | −0.05(0.03) | −0.111 | −0.004 |
| Threat | 0.00(0.01) | −0.015 | 0.003 | 0.01(0.01) | −0.038 | 0.018 | −0.01(0.02) | −0.046 | 0.020 |
| Infection | 0.00(0.01) | −0.011 | 0.010 | 0.00(0.01) | −0.010 | 0.025 | 0.00(0.01) | −0.011 | 0.022 |
| Measures | 0.00(0.01) | −0.011 | 0.024 | 0.00(0.02) | −0.040 | 0.027 | 0.00(0.02) | −0.029 | −0.040 |

The conditional indirect effects of hope, social anxiety and social avoidance are displayed in Table 5. For the sake of readability, we only present significant values (i.e., information), that is, the parameters for which the lower and upper levels of bootstrap 95% CI do not include zero. Results show that the conditional indirect effect of hope on life satisfaction through information was significant for mean levels (a1b1 = 0.029, SE = 0.013; LL = 0.008, UL = 0.057) and high levels of positive affect (a1b1 = 0.023, SE = 0.016; LL = 0.001, UL = 0.062), as well as for mean levels (a1b1 = 0.029, SE = 0.013; LL = 0.008, UL = 0.059) and low levels of negative affect (a1b1 = 0.048, SE = 0.025; LL = 0.008, UL = 0.105). The results from the social anxiety model show a significant conditional indirect effect

through information for mean levels (a1b1 = −0.058, SE = 0.028; LL = 1.121, UL = −0.012) and high levels of positive affect (a1b1 = −0.066, SE = 0.041; LL = −0.158, UL = −0.003), and for mean levels (a1b1 = −0.055, SE = 0.027; LL = −0.117, UL = −0.011) and low levels of negative affect (a1b1 = −0.106, SE = 0.057; LL = −0.237, UL = −0.014). Within the social avoidance model, the indirect effect through information was only significant for mean levels of positive affect (a1b1 = −0.049, SE = 0.029; LL = −0.143, UL = −0.007). To be precise, the indirect effect of hope on life satisfaction through information seems to increase with higher positive affect, whereas the indirect effect of social anxiety seems to decrease with higher positive affect. Because no significant conditional indirect effects were found for threat, infection, and measures, Hypotheses 3 and 4 were only partially supported, which suggests that different levels of positive and negative affect conditioned the effect of information on the relationship between hope–life satisfaction and social phobia–life satisfaction.

**Table 5.** Conditional indirect effects of hope on life satisfaction via information at different levels of positive and negative affect.

| | Mediation Analysis | | | | | | | | |
| | Outcome | | | | | | | | |
| | Hope | | | Social Anxiety | | | Social Avoidance | | |
| Moderator | Estimate (SE) | 95% CI | | Estimate | 95% CI | | Estimate | 95% CI | |
| | | LL | UL | | LL | UL | | LL | UL |
| Positive Affect | | | | | | | | | |
| Low | 0.04(0.02) | −0.002 | 0.084 | −0.04(0.04) | −0.123 | 0.016 | −0.05(0.04) | −0.142 | 0.007 |
| Mean | 0.03(0.01) | 0.008 | 0.057 | −0.06(0.03) | −0.122 | −0.013 | −0.05(0.03) | −0.114 | −0.002 |
| High | 0.02(0.02) | 0.000 | 0.062 | −0.07(0.04) | −0.167 | −0.004 | −0.05(0.04) | −0.144 | 0.015 |
| F | 6.55 *** | | | | | | | | |
| $R^2$ | 0.04 | | | | | | | | |
| Negative Affect | | | | | | | | | |
| Low | 0.05(0.03) | 0.009 | 0.107 | −0.11(0.06) | −0.237 | −0.011 | −0.06(0.05) | −0.179 | 0.033 |
| Mean | 0.03(0.01) | 0.008 | 0.057 | −0.06(0.03) | −0.119 | −0.009 | −0.04(0.03) | −0.104 | 0.003 |
| High | 0.02(0.02) | −0.009 | 0.049 | −0.02(0.03) | −0.085 | 0.016 | −0.03(0.03) | −0.103 | 0.008 |
| F | 7.09 *** | | | | | | | | |
| $R^2$ | 0.04 | | | | | | | | |

*** $p < 0.001$.

## 4. Discussion

In the present study, we examined reactions to the COVID-19 outbreak and its impact on SWB in a Spanish sample. Drawing on research suggesting that evaluations about one's life conditions can influence SWB [37], and that social phobia and hope predict life satisfaction [16,38], we examined whether these relationships could be mediated by individuals' appraisals of COVID-19 depending on different levels of positive and negative affect. Our findings indicated that, among the COVID-19 questions, only the information that individuals reported to know about regarding the situation had an indirect effect on the relationship between hope and life satisfaction and between social anxiety and life satisfaction, which was conditioned on high positive affect and low negative affect. Given the relevance of official public campaigns to inform the population, healthcare policies aimed at mitigating the psychological impact of pandemics should consider the present findings. The remaining of the section addresses the significance of the results.

We found that life satisfaction was predicted positively by hope and negatively by social phobia, which is in line with previous studies showing that hopeful individuals tend to evaluate stressful situations as challenging rather than as threatening, helping them to cope better and increasing their SWB [15,38–40]. Hope about overcoming the pandemic can be related to the unprecedented measures implemented by the Spanish government and the informational publicity that is spread through official and local channels, which could improve people's confidence about winning the battle against

COVID-19. Recent studies have indeed indicated that individuals with more knowledge about COVID-19 were more likely to engage in protective measures to prevent infection [13]. On the other hand, individuals with social anxiety have tended to evoke more negative memories, dwell on negative thoughts, and engage in maladaptive coping strategies, which, altogether, have led them to avoid stressful situations [41]. According to our results, holding a hopeful expectancy in times of the COVID-19 outbreak may contribute to greater life satisfaction, whereas a persistent fear and avoidance of situations that involve social connections seem to diminish it. Considering the context in which participants responded to the questions, those who embraced a positive outlook and expected to overcome the outbreak were more psychologically adjusted, which suggests that hope may serve well to help them deal better with COVID-19 and the emotions associated with it. By contrast, fearing and avoiding actions in public locations may prevent individuals from freely unfolding their routine, and the emotions elicited under such circumstances may have a negative impact on their own SWB.

Despite the fact that research on social phobia has commonly come from a psychiatric disorder perspective [19], the unprecedented nature of the COVID-19 outbreak may have sparked new emotional and cognitive responses—that is, fear, anxiety, and beliefs of social disconnection and distance when out in public [42]. Our study provides evidence in favor of this notion, in that individuals (which they did even at initial stages of the outbreak) can feel prompted to lessen the range of social connections and experience increased anxiety when out in public. Indeed, the main reasons that prompted people to ask for professional help during the COVID-19 outbreak were anxiety and fear of contagion [43]. They may have seen activities that involved human-to-human contact, such as going to a supermarket or a bar, or attending social meetings, as sources of potential danger that needed to be avoided. We presume that not only the fear of social interactions but also the emotions associated with considering the dangers that these situations will entail (to the individual and their loved ones) can explain the negative effects on life satisfaction.

A relevant finding of our research was that the mediating role of the self-assessed degree of information about COVID-19 on the hope–life satisfaction and social anxiety–life satisfaction relationships depended on the participants' having high positive affect. In other words, the mechanisms by which information influenced the relationships of hope and social anxiety with life satisfaction were characterized by high positive affect and low negative affect. This informs us about the conditions under which the amount of knowledge mediated the effect of hope and social anxiety on life satisfaction, which suggests that affect can act as a protective mechanism that boosts the positive effect of hope and buffers the negative effect of social anxiety. The mediation process of being well-informed about COVID-19 (or perceiving to be such) may vary across different levels of affect. High levels of positive affect and low levels of negative affect seem to moderate the indirect effect of hope on life satisfaction—that is, feeling satisfied with life during the outbreak may be the result of experiencing a preponderance of positive over negative emotions that, indeed, would instill individuals the perception of being well-informed and therefore elicit a brighter expectation from the situation.

These findings build upon prior research as they address the effect of emotions on thinking and decision-making. Affect is important in decision-making because people tend to seek information that is congruent with their own mood and preferences [44], and in doing so they experience psychological gains, such as less negative affect and more optimism [45]. Positive affect facilitates problem-solving and decision-making, presumably as a result of applying more flexible information-processing strategies that integrate all the information, or almost all [44]. Accordingly, individuals with high positive affect may see the "general picture" and display strategies to search sufficient information, while they avoid focusing on the problem and on information that precludes preventive behaviors. In contrast, negative affect is related to more biased judgments [46] and narrowed cognitive and behavioral systems, and thus it leads to maintain problem-focused thoughts and attentional fixation in the face of threats [39]. Given the negative emotional reactions evoked by the COVID-19 outbreak [43], people with greater concerns and negative emotions at initial stages may be more prone to seek for information.

Similar to the reaction during the SARS pandemic, one of the strategies the Spanish government adopted during the COVID-19 outbreak was to launch publicity campaigns to inform citizens about the pandemic. Providing more information about the focus of the emergency (e.g., the virus), including dissemination of information about precautionary measures and avoiding crowded areas, helped people to cope with previous outbreaks [47]. Ambiguous information or the absence thereof can instill in individuals a sense of uncontrollability, which can increase their anxiety [48]. Recent studies on COVID-19 have shown that a large amount of uncontrolled news was rapidly spreading through the Internet, with the risk of fake news proliferating faster than the virus itself, thereby generating uncertainties and worries [7]. Constantly reading negative news or trusting ineffective communication can have negative consequences on well-being [48,49], and this detrimental consequence could be accentuated by the negative emotions experienced due to COVID-19.

*Limitations*

This study has certain limitations to be noted. First, although the time of data collection can provide valuable information about the early stages of the outbreak, the sample comprised Spanish individuals with access to the Internet and to social media. This is an important limitation since participants were mostly women, employed and graduated, and who could be more informed about the outbreak and be more willing to engage in our research. Because the sample was non-representative, interpretations of the findings are subjected to socio-cultural constraints, and thus caution must be applied when interpreting the results. In order to provide valuable data about the costs of COVID-19 to mental health, research should avoid non-probabilistic samples and reach under-represented populations, especially those mentally vulnerable to the pandemic [50]. Cross-cultural studies can investigate whether our results are replicated in different countries during the COVID-19 crisis. Second, the cross-sectional nature prevented the generalization of the results, and it did not allow us to study the causality of the studied relationships. The limited time in which the survey was developed could indicate inadequacy in the evaluation of the variables. Furthermore, questions about COVID-19 may not comprehensively capture individuals' responses to the outbreak; for instance, we could not infer how the respondents obtained the information (official channels, experts, television, social media, acquaintances, etc.). More specific questions about the COVID-19 responses should be inspected in future studies. In addition, longitudinal studies to analyze the dynamics of SWB during COVID-19 can provide valuable insights into understanding how the relationship between the study variables unfolds. Third, the use of self-reports does not allow for the drawing of conclusions about causality; therefore, future studies should adopt experimental designs to verify the findings of the current study.

**Author Contributions:** "Conceptualization, A.B.-B. and C.A.; methodology, A.B.-B.; formal analysis, A.B.-B.; investigation, A.B.-B., C.T.-G., C.T.-N. and C.A.; data curation, A.B.-B.; writing—original draft preparation, A.B.-B., C.T.-G. and C.T.-N.; writing—review and editing, A.B.-B. and C.T.-G.; supervision, C.A.; funding acquisition, C.A. All authors have read and agreed to the published version of the manuscript.

**Funding:** "This research received no external funding."

**Conflicts of Interest:** "The authors declare no conflict of interest."

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
