# Peer review of "The Costs of the COVID-19 on Subjective Well-Being: An Analysis of the Outbreak in Spain"

_sustainability, doi:10.3390/su12156243_

Round 1
Reviewer 1 Report
Thank you for taking into account my comments. I am a bit surprised that you had not separated answers received before and after the lock-down policy in your first analysis though.
Author Response
Thank you very much for your contributions to improve the study.

Reviewer 2 Report
First of all, I thank the authors for their careful consideration of the reviewers' comments and the detailed covert letter they facilitated. In my previous review, my concerns focused mainly on methodological aspects. From my point of view, the changes introduced by the authors in this direction have contributed to improving the clarity and understanding of the manuscript. However, I think there is still room for improvement in its expository quality, as I indicate below.
MATERIALS AND METHODS
I believe that some of the explanations that the authors have provided in the cover letter should be included in the manuscript, especially regarding the justification of the representativeness of the sample analyzed. For example, they should refer to the advantages of using bootstrapping and the number of boostrap samples they have worked with. In any case, taking into account the small number of surveys collected in provinces other than Catalonia, I believe that the authors should consider whether the results can be extrapolated only to Catalonia (and not to the country as a whole).
INSTRUMENTS
In order to get a better placement of the reader, I would suggest for instruments to be introduced in the manuscript in the same order as they are described in Table 1. That is to say, firstly the Life Satisfaction as dependent variable or criterion, secondly the predictors, next the two moderators and, finally, the mediators.
DATA ANALYSIS AND RESULTS
* Not all readers who perform mediation analysis use the same macro (PROCESS with SPSS) as the authors. Therefore, the authors should provide the necessary statistical information, both in the specification of the models and in the reporting of the results so that any reader can understand the results and assess their reliability. In this vein, I strongly recommend that in addition to Figures 1 and 2, where conceptual diagrams are displayed, the equations to be estimated should be collected, with the names of the coefficients, a1 , b1, c1, c1 ', etc., as it is done in the methodological articles cited by the authors (e.g. Hayes 2015). For the sake of brevity, they should introduce this information just for "hope" and "information", that is H1 and H3. I also recommend indicating in the results what tests they performed to analyze whether there is an indirect mediation effect and how the results should be interpreted (as the authors have responded in the cover letter to comments 4 and 6 of my review). That is, simply indicate that they contrast if the indirect effect (product of coefficients or mediation index) is statistically different from zero, estimating the upper level (UL) and lower level (LL) of the confidence interval (if zero is not included, the indirect effect is statistically significant). In the current version of the manuscript it is very hard to understand the results.
* I do not understand what does "total indirect effect (β = .031)" mean (line 224).
* I reiterate in my comment on the previous review. From line 224, the reporting of the results is not understandable. When parameters are estimated in a regression model, they are generically called B, but when there are so many estimated parameters, it is very convenient to name them specifically. In this type of analysis the coefficients or paths already have a denomination (a1, b1, etc.) as it is shown in the Figures 1 and 2, in which the indirect effects are expressed as products of those coefficients (for example a1b1). In addition, in some models the interaction between the dependent variable and the moderating variable is considered. For all these reasons, I consider key that authors include the equations of the models whose parameters are estimated. Furthermore, the results of indirect effects in the simple models of mediation (estimated values and confidence intervals) might be reported in a Table (similar to Table 4).
* The numbering of sections 3.2 and 3.3 should be reviewed.
DISCUSSION
Taking into account the results reported in Table 4, where the confident interval for positive affect hope-life satisfaction is (0.000, 0.062), some of the ideas collected should be reviewed. For example:
* Lines 281-284, "Our findings indicated that, among the COVID-19 questions, only the amount of information they knew about the situation had an indirect effect on the relationship between hope and life satisfaction, which was conditioned on a preponderance of positive affect (high positive affect and low negative affect)".
* Lines 323-325, "That is, the mechanisms by which information influenced the relationships of hope and social anxiety with life satisfaction were characterized by high positive affect".
Reviewer 3 Report
This is, in summary, an interesting paper aimed to examine reactions to the COVID-19 outbreak and the mediating role of these reactions in the associations of hope and social phobia with life satisfaction through different levels of positive and negative affect in a sample of 541 Spanish 9 adults. The authors reported that, based on moderated mediation analysis, the conditional indirect effect of hope and social anxiety on life satisfaction through information depended on the participants’ having high positive affect. In addition, affect seems to be a mechanism able to modulate the influence of individuals’ perception about the COVID-19 on their life satisfaction. Moreover, those with high positive affect might see the “general picture” and search for adequate information as they avoid focusing on the problem and on specific information that precludes preventive behaviors. Finally, having a positive affect might help individuals to adopt information-processing strategies during the COVID-19 outbreak that will improve their life satisfaction.
The authors may find as follows my main comments/suggestions.
First, when throughout the Introduction section, the authors correctly reported that subjective well-being is considered an indicator of mental health and the experience of negative affect also appeared to be common, they could even briefly mention the possible association between subjective well-being, anxiety/depression, lack of motivation and alexithymic traits during pandemic. In particular, there is an increasing evidence that psychological distress together with alexithymic traits may be considered risk factors for negative outcomes (e.g., suicide), even simply increasing the risk of development of depressive symptoms or per se. In order to briefly discuss this topic (although i understand that the link between anxiety/depression and alexithymic traits and suicidal behavior is not the main topic of the present manuscript), i suggest to cite, within the main text, the paper published in 2017 on Frontiers in Psychiatry (PMID: 28855878). In addition, the involvement of sensory perception in emotional processes and clinical outcomes has been well demonstrated. Importantly, the unique sensory processing patterns of individuals with major affective disorders and their relationship with psychiatric symptomatology have been clearly reported. Hyposensitivity or hypersensitivity may be "trait" markers of individuals with major affective disorders and interventions should refer to the individual unique sensory profiles and their behavioral and functional impact in the context of real life. Thus, given the above information, my additional suggestion is also to include throughout the manuscript, the paper published in 2016 on Psychiatry Res (PMID: 26738981).
Moreover, the most relevant psychometric instruments could be described more succinctly.
In addition, whether the local review board regulalry approved the present study design is a matter of debate and needs to be clarified for the general readership.
Furthermore, the authors should immediately present and discuss, in the first lines of the Discussion section, their most relevant study findings. Conversely, they seem to focus on the main aims of this paper that should have been stressed elsewhere.
Finally, what is the take-home message of this paper? While the authors concluded that having a positive affect might help subjects to adopt information-processing strategies during the COVID-19 outbreak that will improve their life satisfaction, they failed, in my opinion, to provide some conclusive remarks of their paper. Here, some additional details/information are needed.
Round 2
Reviewer 3 Report
In the revised manuscript, the authors addressed sufficiently most of the major questions raised by Reviewers.
This manuscript is a resubmission of an earlier submission. The following is a list of the peer review reports and author responses from that submission.
Round 1
Reviewer 1 Report
Being a social scientist but not a social psychologist, I am not at ease evaluating this particular paper. A number of paradigmatic differences hinder my capacity to compare the paper to standard procedures in social psychology. This has to be discounted from my mostly critical judgments below.
The paper relies on data from a snowball sample whose characteristics appear to be quite ad hoc and certainly not representative of the population. This poses a major problem to the generalizability of results, unless we assume that psychological processes are independent of social, economic and cultural characteristics. Psychologists sometime hold this view, which is however hard to buy in the rest of the social sciences. While the authors acknowledge this limitation, such an acknowledgment does not exempt from doubts about the results.
Moreover, the timing of data collection – before the lockdown – is not ideal, as information about the seriousness of the situation was not evenly distributed in the population.
Another major problem of the paper is that its hypotheses are not ‘if then’ statements, but rather very general and vague assertions about the mediating role of some psychological scale between two other psychometric constructs. Now, ‘mediation’ can mean several things (amplification or reduction of effects), and is quite common that psychometric constructs are somewhat correlated one another.
The results are also not entirely convincing. On the one hand, the paramount finding – more (self-assessed) information about Covid reduces negative affect and tightens the association between ‘hope’ and ‘life satisfaction’ – is unlikely to be specific to a pandemic situation. On the other hand, the study has no control on the factual information of people (eg, from which sources? How deep and sound is individual information?).
All in all, there is an enormous amount of endogeneity in the analysis, and there is no way to discern whether respondents were ‘treated’ differently. Again, the authors themselves are aware of this limitation in the final section, but this is not by itself a remedy.
Reviewer 2 Report
This is a paper on a timely topic, i.e., associations between reactions to the COVID-19 outbreak and subjective well-being in Spain. The paper has some serious limitations, however, which will be outlined in the following:
1) The rationale for the selection and use of constructs is not clearly outlined in the introduction.
Subjective well-being entails a cognitive component, life satisfaction, and an affective component, positive and negative affect (Diener, 1984). However, in the paper only life satisfaction is considered to be an outcome/indicator of subjective well-being, whereas positive and negative affect are considered to be moderator variables. The rationale for this approach is not clear. As stated correctly, "emotional experiences can influence life satisfaction statements" (l.34), however, this would imply that there is a main effect of affect on life satisfaction, but it does not justify the moderator effect examined.
Also, although a reduction of social connections was necessary to slow down the spread of the virus, I am not convinced this is captured adequately by examining social phobia, which also entails fear and avoidance of performance situations. As the correlation table shows, social phobia was not related to fear of infection as the authors imply in the introduction (l.98-99).
Hope seems to be a more adequate predictor of subjective wellbeing, also during a pandemic, however, it is not clearly outlined in the introduction why hope is assumed to be associated with the COVID-19 evaluation items, which would be necessary to establish the mediator assumption.
2) The authors confuse moderation and mediation. For example, in the hypotheses H3 and H4 it is stated "...will indirectly influence the relationship between hope / social phobia and life satisfaction through positive and negative affect". An indirect effect signifies a mediator effect, however, and not the moderator effect that was examined apparently. Also, in the discussion the authors state "...through low levels of negative emotions" (l. 316), which, again, would signify a mediator effect of affect but does not adequately describe a moderator effect. Overall, the moderator effects are not adequately explained in the discussion section.
3) Not surprisingly given the very limited associations between predictors and mediators as well as mediators and outcome (see Table 1), most indirect effects were not significant in the mediator analyses. However, the authors state that "these results provided support to hypothesis 1 and 2" (l.234-235). This claim does not seem to be justified.
4) The study is cross-sectional, which is a serious limitation for drawing conclusions about causality and examining mediation processes in particular. This is not outlined adequately in the discussion section. In the method section, the authors even imply a reverse direction of causality (i.e., "the influence of stable judgements of life satisfaction on social phobia and hope", l. 189).
5) The fact that the sample is not representative limits the generalization of the results. This is not outlined adequately in the discussion section.
Reviewer 3 Report
The subject of the article is of great interest and topicality. The literature review and structure are appropriate and adequate. However, my main concerns are related to the methodological aspects: the sample and the statistical models carried out.
Major aspects
* Regarding the sample, firstly, it is well known the numerous bias problems of using the snowball sampling method. Secondly, I wonder if the small sample size (N=562) could be considered representative of the whole country. Some statistical justification in this way should be presented. Thirdly, no information regarding the geographical representativeness of the whole country is referred. Where was the survey carried out? (In which different provinces, for instance.)
* When instruments are introduced, the Cronbach's Alpha is usually reported.
* I have serious doubts about the strength of the empirical strategy followed in the paper and, therefore, about the reliability and robustness of the results. I will focus on variables Life Satisfaction (dependent variable), hope (independent variable) and information (mediator variable), since it is the only case were statistically significant relationships have been found. Let consider regression (1) between the independent and dependent variable, regression (2) between the mediating variable and the independent variable, and regression (3) between the independent variable and dependent variable after adjusting the effect of the mediating variable. For mediation to take place, one should observe that both the coefficient (c1') and significance level of the independent variable (hope) in regression (3) substantially diminish in comparison to those obtained (c1) in regression (1). This fact signals that some of the effects of the hope on life-satisfaction might be mediated via information. Otherwise, if the mediator variable does not substantially alter the relation of independent variable to dependent variable, we would have an example of a covariate. In order to check that coefficient c1' is substantially lower than c1, difference in coefficients test might be performed.
* In addition, the rationale for estimating moderated mediation models is that the effect of hope on Life Satisfaction via COVID_19 variables differs depending on the levels of positive affect (or negative affect). That is, the main assumption is that there is a hierarchical or nested structure determined by the three levels of positive (or negative) affect. Therefore, the first step of the moderated mediation analysis should be to empirically test if the independent variables show statistically significant differences between the levels of positive (or negative) affect, calculating, for example, the Intraclass Correlation Index.
Minor aspects
* The wording of the Introduction Section should be revised to avoid repeating the same ideas in different parts. I consider that Section 1.1 should be the first paragraph of the paper (it is the most general). In addition, I believe that the first paragraph of Section Introduction (lines 22-40) is a summary of the rest of sections (1.1, 1.2, 1.3 and 1.4), thus I would suggest to delete the repeated ideas, because there is already an abstract of the paper.
* In order to get a better understanding of the results of the regressions, it would be convenient to first write the regression equations and then always use the same notation to refer to the parameters of the regressions. The use of notation "a, b and c" and also notation "β" is confusing.
* The last sentence (Section 2.1, lines 143-143) is not correct, because the state of alarm in Spain was declared on March 14 ("Data was collected from 12 to 19 of March 2020, a week before the Spanish government declared the state of alarm and the full restrictions were implemented").
* Some references are not quoted in the References Section (for instance MacKinnon, Lockwood, Hoffman, West, & Sheets, 2002).
Reviewer 4 Report
Thank you for the opportunity to review this interesting manuscript. It is generally well-written and pleasurable to read. A strong point is the consistent use of validated measurement scales for which we have the info and reference on their validation in the Spanish language, which is a rigorous point. I, however, have some suggestions for improving the current version of the manuscript.
- Introduction
The definition of subjective well-being is very restrictive/reductive: “subjective well-being (SWB), namely their degree of information or accordance with the government measures.”. I would suggest to adopt a somewhat broader (and more accurate) definition.
“At that point, the immediate socio-psychological effects of this outbreak on individuals’ SWB has started to consolidate evidences”: the sentence is unclear, I think the subject of the verb ‘consolidate’ is incorrect, please rephrase.
In the section “The Coronavirus Pandemic: A contextualisation”, I think it would be good to add more info on the policy measures developed in Spain (the specific context of the study) as they can have an important role in changing people’s emotions and perceptions of the epidemic. If you need help in describing these measures, see for instance: https://www.covid19healthsystem.org/countries/spain/countrypage.aspx
“Besides, the rapid increase of cases in a short-period time drove the WHO to take restrictive measures that are carrying worldwide psychological consequences (Wang et al., 2020).”: It is not the WHO which took the main restrictive measures but national governments (some of them have adopted completely different measures). The WHO just has an advisory role. Please edit accordingly.
I think it could be interesting to discuss a bit more the effect of the lock-down (not only the pandemic itself) on mental health affects – especially as the lock-down was very strict in Spain - might be interesting to read and quote the following paper: Brooks et al. The psychological impact of quarantine and how to reduce it: rapid review of the evidence. Lancet. February 2020. Except if the study was carried out before the implementation of the lock-down in Spain (then skip this comment), but we do not have this info anywhere in the paper, please add it somewhere.
“and whether it dependent on the levels of positive and negative affect.” : is a word missing maybe “was dependent”?
- Materials and methods
Participants and procedures
We need more details on how the snowball sampling method was carried out; how were the initial participants selected and contacted? How was the link of the study disseminated in the first step? Through social media? This needs to be told, this is a key information.
Did you collect any identifying information such as e-mail addresses?
The description of participants’ characteristics should be in the result section.
The date of the questionnaire should be more closely linked to the dates of the lock-down measures or other key measures implemented in Spain. We need to know if it’s before or after the lock-down at least (because the social phobia scale might make no sense if the lock-down measures were already implemented).
What was approximately the duration of the online survey? 20 minutes? Please add it.
Instruments
“It is a 12-item questionnaire measuring ... that participants respond using an 8-point Likert scale”: I don’t understand this sentence, is a word missing maybe?
“(1) Which is your degree of information related to the COVID-19?”: replace “which” by “what”?
Results
Figure 1, and accompanying text, would be more adequately located in the methods section. The same applies to Figure 2.
Discussion
There seems to be an over-representation of women (who often suffers from more negative mental health affects) and people with high education level (who are often more informed, more involved in their health, and more aware of governmental measures) in your sample. We need to be able to compare it to the percentages in the general population (add them here or in the result section when you describe participants characteristic). This absolutely needs to be discussed more in the limits of your study, especially the impact it could have on your findings. How confident are you that your results are informative for the Spanish population? See: https://www.thelancet.com/journals/lanpsy/article/PIIS2215-0366(20)30237-6/fulltext. This needs to be discussed much more in your paper, especially as this is the key limit of the study (and cannot be overlooked) and as your sample size is really small for an online survey.
“negative consequences to well-being” : “on” well-being
“An economic model elaborated on the costs and benefits of ignoring and sustaining positive affect at societal levels (see Hermalin & Isen, 2008).”: I don’t understand this sentence, maybe a verb is missing?
“Nurturing feelings of personal worthiness or helpfulness through the dissemination of information may be valuable assets address health crisis all at once.”: add “to” before “address”.
“The majority of the literature related to infectious disease agree to use the proper channels for providing accurate information, opting for one-way transmission of information through expert sources and often via the mass media (Holmes, 2008).”: infectious diseases: plural form; agrees: with a s. Maybe an in-depth proof-reading of such typos might be useful.
Others
Why are author contributions, funding and conflicts of interest empty? Please fill in these sections.